# S2P: State-conditioned Image Synthesis for Data Augmentation in Offline Reinforcement Learning

**Daesol Cho**[*]
Seoul National University
Automation and Systems Research Institute (ASRI)

**Dongseok Shim**[*]
Seoul National University
Interdisciplinary Program in AI

**H. Jin Kim**
Seoul National University
Artificial Intelligence Institute of Seoul National University (AIIS)
{dscho1234, tlaehdtjr01, hjinkim}@snu.ac.kr

## Abstract

Offline reinforcement learning (Offline RL) suffers from the innate distributional shift as it cannot interact with the physical environment during training. To alleviate such limitation, state-based offline RL leverages a learned dynamics model from the logged experience and augments the predicted state transition to extend the data distribution. For exploiting such benefit also on the image-based RL, we firstly propose a generative model, S2P (State2Pixel), which synthesizes the raw pixel of the agent from its corresponding state. It enables bridging the gap between the state and the image domain in RL algorithms, and virtually exploring unseen image distribution via model-based transition in the state space. Through experiments, we confirm that our S2P-based image synthesis not only improves the image-based offline RL performance but also shows powerful generalization capability on unseen tasks.

## 1 Introduction

Deep learning algorithms have shown significant development thanks to the large pre-collected dataset, such as SQuAD [47] in natural language processing (NLP), and ImageNet [4] in computer vision. On the contrary, reinforcement learning (RL) requires an online trial-and-error in training process to collect the data by interacting with the environment, which hinders its utilization in many real-world applications. Due to this intrinsic property of current online RL algorithms, there exist some approaches that try to deploy large and diverse pre-recorded datasets without online interaction with the environment, which is called offline RL.

However, recent studies have observed that the current online RL algorithms [10, 35] perform poorly in an offline setting. It is primarily attributed to the large extrapolation error when the Q-function is evaluated on out-of-distribution actions, which is called the distributional shift [25, 22, 8]. That is, due to the offline

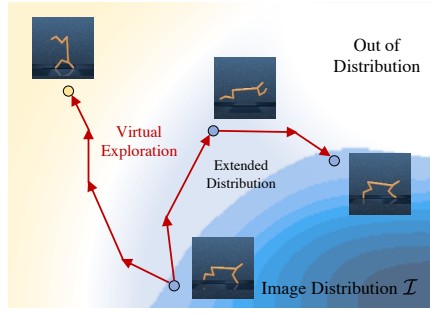

Figure 1: S2P generates the dynamics-consistent image transition data by virtually exploring in the state space to extend the distribution of the offline datasets.

---

[*]Equal contribution

36th Conference on Neural Information Processing Systems (NeurIPS 2022).

setting that limits online data collection, the offline RL has struggled to generalize beyond the given offline dataset. Even though some offline RL methods [26, 24, 54] achieve reasonable performances in some settings, their training is still limited to behaviors within the given offline dataset distribution, and detours the evaluation on out-of-distribution data rather than directly addressing such an empty space of the offline dataset. Thus, there exists a growing need for the development of algorithms specialized to directly address such out-of-distribution data by extending the offline dataset distribution's support.

To alleviate the fixed dataset distribution problem, recent studies propose some data augmentation strategies. In state-based RL, model-based algorithms [2, 1, 27, 16] which learn a dynamics model from the pre-recorded dataset and augment the dataset with generated state transitions have emerged as a promising paradigm. As the model-based approach trains dynamics models in a supervised manner, it allows a stable training process and generates reliable state transition data for augmentation. Thus, it can be a plausible choice that it enables the generalization into the unseen state-action by performing dynamics-consistent planning on unseen state distribution.

When it comes to the image domain, however, there is still no augmentation strategy to mitigate the aforementioned distribution shift. Even though some model-based image RL methods [12, 11] that propose to learn latent dynamics using reconstruction error from ELBO objective [18, 51, 23] can be exploited for generating image transition data in a similar manner to the state-based methods, the quality of the output images from these approaches is not satisfactory because 1) their focus is on learning latent representation suitable for the RL network's inputs rather than generating high-quality and accurate images, and 2) ELBO-based objective cannot generate photo-realistic images compared to other generative models such as GAN [9] or diffusion [14, 41] and it usually produces blurry outputs. Above all, 3) model-based image RL only exploits image input and it makes the generative model fail to capture the accurate dynamics and the details of the agent's posture or objects in the image [40, 42, 3]. These undesired properties discourage offline RL algorithms from adding the reconstruction output of model-based image RL to their training data as an augmentation strategy.

Therefore, we propose S2P (State2Pixel) which utilizes multi-modal input (the state, and the previous image observation of the agent) to synthesize the realistic image from its corresponding state. The key element of S2P is a multi-modal affine transformation (MAT) which effectively exploits both state and image cross-modality information. Unlike previous learned affine transformation [19–21, 44, 34], which leverages a single domain input, MAT fuses the cross-modal representation from the state and the image to produce the scale and the bias modulation parameters. This multi-modality of S2P makes it possible to generate the dynamics-consistent images from the reliable state transition while preserving high-quality image generation capability.

To sum up, our work makes the following key contributions.

- We propose a state-to-pixel generative model (S2P) which generates dynamics-consistent image and multimodal affine transformation (MAT) module for aggregating cross-modal inputs.

- To the best of the author's knowledge, this work is the first to propose image augmentation for offline image RL and overcome innate fixed distribution problem by implicitly leveraging reliable state transition.

- We evaluate our S2P on the DMControl [52] benchmark environments with the offline setting, and it results in $1.0 - 3.0$x higher offline RL performance by augmenting the generated synthetic image transition data.

- Even with the state distributions of the unseen tasks, S2P can generalize to unseen image distribution, and we show that the agent can be trained by offline RL with these generated images only.

## 2   Related Work

### 2.1   Image Synthesis

Generative Adversarial Networks (GAN) [9] based deep generative models enjoy huge success in synthesizing high-resolution photo-realistic images via style mapping function and the learned affine transformation [19–21, 44, 34]. StyleGAN [19] firstly utilizes a style vector $\mathbf{w}$ and Adaptive Instance

Normalization (AdaIN) [5, 15] in the generative networks to disentangle the latent space and control the scale-specific synthesis. Following studies [20] pinpoint that the AdaIN operation which leads to information loss in the feature magnitude makes undesired droplet-like artifacts in the synthesized images and proposes weight demodulation by assuming the variance of input features. SPADE [44] proposes an architecture to synthesize the image using its corresponding semantic masks and spatially learned affine transformation. ManiGAN [34] suggests Text-Image Affine Combination Module (ACM) which enables the network to manipulate the images using text descriptions given by users. The difference between our proposed MAT and the previous studies is that we leverage cross-modal data, state and image, to estimate the modulation parameters for the learned affine transformation whereas other studies use a single data type such as text or image.

## 2.2  Offline Reinforcement Learning & Data Augmentation

Offline RL [6, 29, 33] is the task of learning policies from a given static dataset, which is different from online RL that learns useful behaviors through trial-and-error in the environment. Prior offline RL algorithms are designed to constrain the policy to the behavior policy used for offline data collection via direct state or action constraints [8, 37], maximum mean discrepancy [25], KL divergence [54, 59, 17], or learning conservative critics [26, 24]. However, most of these methods are limited to exploiting the state-action distribution of the given static dataset, rather than exploring and extending the distribution. As the offline setting prohibits online interaction with the environment, we suggest the synthetic data generation method to enable the offline RL agent to virtually explore and extend the distribution by leaving the support of the dataset.

Recent works in model-based state RL that involves learning a policy with a dynamics model [22, 2, 1, 28, 57, 58] suggest the need to augment the data with generated transitions from the model for extending the data distribution. Also, augmentation strategies in image domain [30, 49, 31, 55, 56] emphasize the importance of image augmentation for sample efficiency and robust representation learning. But, these works focus on purely image manipulation techniques on the given image such as cropping rather than generating image transition. Some image-based methods [45, 11, 12] that use the variational model to train the latent dynamics are studied in a similar concept to the state-based ones. However, the generated image from these methods is the byproduct of learning the effective image representation rather than the main purpose of these works, and as these methods only utilize the image input, it leads to missing objects or inaccurate dynamics of the agent in the reconstructed image. Therefore, to bridge the gap between the model-based state transition data augmentation techniques and image augmentation, we propose the method that generates dynamics-consistent image transition data along timesteps with multi-modal inputs.

## 3  Method

### 3.1  S2P Generator

**Architecture.** The goal of S2P (State2Pixel) generator is to synthesize the image $\hat{I}_t$ which perfectly represents all the information of its corresponding state $s_t$. Unfortunately, a state-sole condition cannot formulate a single deterministic rendered image because, in most cases, state does not provide the agent's position from the global coordinate, but rather from an egocentric coordinate. Also, image-based RL algorithms utilize sequential images to capture the agent's velocity using the change of the background, e.g. ground checkerboard, between input images. It means that the image of the current step $I_t$ is dependent not only on the current state $s_t$, but also on the image of the previous step $I_{t-1}$. We, therefore, build the generator $G$ to synthesize the image $\hat{I}_t$ from both the state $s_t$ and the previous image $I_{t-1}$ so that the generated image $\hat{I}_t$ can preserve the dynamics-consistency in the physical environment.

$$\hat{I}_t = G(s_t, I_{t-1}) \tag{1}$$

At the first layer of S2P, the input image $I_{t-1}$ and the state $s_t$ are projected to the feature space via convolution and MLP encoders respectively. Both features are then fed to the hierarchical generator block which consists of several residual connections [13] and the upsampling layer. After passing through each generator block, the spatial size of the feature map is doubled while the channel dimension is halved. The image features are converted to the RGB images at the last layer of the generator with a single convolution layer.

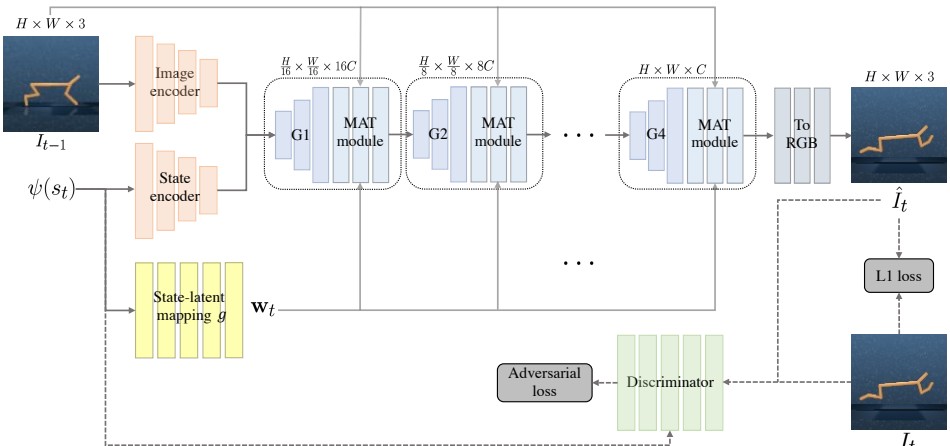

Figure 2: An overview of S2P architecture. State $s_t$ and the previous image $I_{t-1}$ are used as input to generate current step image $\hat{I}_t$. The spatial size of the features gets larger as it passes through multiple upsampling generators. G and MAT indicate the generator block and the Multimodal Affine Transformation respectively

.

We observe that the input signals, i.e. state $s_t$ and image $I_{t-1}$, become attenuated as they pass through deeper generating layers and the network produces images with poor quality. So, similar to recent style-based image synthesis algorithms [19–21, 44, 34], we adopt a learned affine transformation architecture to inject auxiliary signals to the generator. The difference between the previous style-based generative models and S2P is that we propose a multimodal affine transformation (MAT) to produce the learnable modulation parameters, $\gamma$ and $\beta$, with the cross-modality representation via a multimodal feature extractor and state-to-latent mapping function. The overall architecture of our proposed S2P is depicted in Figure 2.

A non-linear latent mapping function $g : \mathcal{S} \rightarrow \mathcal{W}$ which is implemented as an 8-layer MLP produces a latent code $\mathbf{w} \in \mathcal{W}$ from the given state $s$ in the state space $\mathcal{S}$. The latent code $\mathbf{w}$ is spatially expanded as the same size of the input feature of MAT module $\mathbf{x}$, and the conditioned image $I_{t-1}$ is also linearly interpolated to make its size same as $\mathbf{x}$. The spatially expanded $\mathbf{w}$ and the resized $I_{t-1}$ are channel-wisely concatenated and fed to the multimodal feature extractor to fuse the state and image cross-modality representation. Each estimator then produces the learnable scale $\gamma$ and bias $\beta$ for effective cross-modal affine transformation.

Finally, our proposed MAT operation is defined as:

$$\mathbf{y}^i = \gamma^i(\mathbf{w}, I_{t-1}) \odot \frac{\mathbf{x}^i - \mu_c(\mathbf{x}^i)}{\sigma_c(\mathbf{x}^i)} + \beta^i(\mathbf{w}, I_{t-1}), \tag{2}$$

where $\mu_c(\cdot)$ and $\sigma_c(\cdot)$ are the channel-wise mean and the standard deviation of the input feature of MAT $\mathbf{x}^i$ from the $i^{th}$ block of the generator, and $\odot$ denotes Hadamard element-wise product. The design of MAT is illustrated in Figure 3.

In addition, it is shown that the neural network is biased toward learning a low frequency mapping and has difficulty in representing a high frequency information [46, 39]. To mitigate such undesired tendency, we do not use naïve state vector $s$ as input, but employ a positional encoding with the high frequency function $\psi : \mathbb{R} \rightarrow \mathbb{R}^{2L}$ which is defined as:

$$\psi(x) = (\sin(2^0 \pi x), \cos(2^0 \pi x), \cdots, \sin(2^{L-1} \pi x), \cos(2^{L-1} \pi x)), \tag{3}$$

where $x$ indicates each component of state vector $s$.

We utilize multi-scale discriminators in [53] to increase the receptive field without deeper layers or larger convolution kernels for alleviating the overfitting. Two discriminators with identical architecture are adopted during training, and the synthesized and real images which are resized to several spatial sizes are fed to each discriminator.

**Loss Function.** Our S2P generator is trained by linearly combined multiple objectives. First, we leverage a pixel-wise $\mathcal{L}_1$ loss between the output of the generator $\hat{I}_t$ and the real image $I_t$,

$$\mathcal{L}_1 = ||I_t - \hat{I}_t||_1. \tag{4}$$

In addition to the pixel-wise loss, we also utilize the ImageNet [48] pre-trained VGG19 [50] network to calculate the perceptual similarity loss $\mathcal{L}_{per}$ between two images,

$$\mathcal{L}_{per} = ||\phi^{(i)}(I_t) - \phi^{(i)}(\hat{I}_t)||_1, \tag{5}$$

where $\phi^{(i)}$ denotes the i-th layer of VGG19.

We implement the adversarial objective $\mathcal{L}_{adv}$ for both S2P generator and multi-scale discriminators same as pix2pixHD [53]. The difference is that we replace the least square loss with the hinge-based loss [36] and we condition state information to the discriminator $D$ so that the generator is induced to produce the dynamics-consistent outputs.

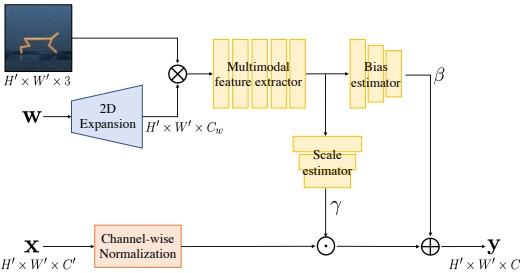

Figure 3: Multimodal Affine Transformation (MAT) module.

$$\mathcal{L}_D = -\mathbb{E}[\min(0, -1 + D(s_t, I_t))] - \mathbb{E}[\min(0, -1 - D(s_t, \hat{I}_t))], \ \mathcal{L}_G = -\mathbb{E}[-D(s_t, \hat{I}_t)]$$
$$\mathcal{L}_{adv} = \mathcal{L}_G + \mathcal{L}_D \tag{6}$$

The total loss function to optimize the S2P generator can be defined as:

$$\mathcal{L} = \lambda_1 \mathcal{L}_1 + \lambda_2 \mathcal{L}_{per} + \lambda_3 \mathcal{L}_{adv}, \tag{7}$$

where $\lambda_1$, $\lambda_2$, and $\lambda_3$ are the hyperparameters to balance among the objectives.

### 3.2 Offline reinforcement learning with synthetic data

We consider the Markov decision process (MDP) $\mathcal{M} = (\mathcal{I}, \mathcal{S}, \mathcal{A}, T, r, \rho_0, \gamma)$, where $\mathcal{I}$ denotes the image space, $\mathcal{S}$ the state space corresponding to $\mathcal{I}$, $\mathcal{A}$ the action space, $T(s'|s, a)$ the transition dynamics, $r(s, a)$ the reward function, $\rho_0$ the initial distribution, and $\gamma$ the discount factor. We denote the discounted image visitation distribution of a policy $\pi$ using $d_{\mathcal{M}}^{\pi}(I) := (1 - \gamma) \sum_{t=0}^{\infty} \gamma^t \mathcal{P}(I_t = I|\pi)$, where $\mathcal{P}(I_t = I|\pi)$ is the probability of reaching image observation $I$ at time $t$ by using $\pi$ in $\mathcal{M}$. Similarly, we denote the image-action visitation distribution with $d_{\mathcal{M}}^{\pi}(I, a) := d_{\mathcal{M}}^{\pi}(I)\pi(a|I)$. The objective of RL is to optimize a policy $\pi(a|I)$ that maximizes the expected discounted return $J(\pi) = \frac{1}{1-\gamma} \mathbb{E}_{(I,a) \sim d_{\mathcal{M}}^{\pi}(I,a)}[r(s, a)]$.

In the offline RL setting, the algorithm has access to a static dataset $D = \{(I_t, s_t, a_t, r_t, I_{t+1}, s_{t+1})\}_{t=0}^{N}$ collected by unknown behavior policy $\pi_\beta$. In other words, the dataset $D$ is obtained from $d_{\mathcal{M}}^{\pi_\beta}(I, a) := d_{\mathcal{M}}^{\pi_\beta}(I)\pi_\beta(a|I)$ and the goal is to find the best possible policy using the static dataset without online interaction with the environment.

To utilize the dynamics-consistent transition data for augmentation in offline RL, we take the model-based approach that trains an ensemble of dynamics and reward model $\hat{T}_\theta(s', r|s, a)$, which outputs the predicted next state, and reward $\hat{r}(s, a)$. Once a model has been learned, we can construct the learned MDP $\widehat{\mathcal{M}} = (\mathcal{I}, \mathcal{S}, \mathcal{A}, \hat{T}, \hat{r}, \rho_0, \gamma)$, which has the same spaces, but uses the learned dynamics and reward function. Naively optimizing the RL objective with the $\widehat{\mathcal{M}}$ is known to fail in the offline RL setting, both in theory and practice [22, 57], due to the distribution shift and model-bias. To overcome these, we take an uncertainty estimation algorithm like bootstrap ensembles [43, 38], and obtain $u(s, a)$, an estimate of uncertainty in dynamics. Then we could utilize the uncertainty penalized reward $\tilde{r}(s, a) = \hat{r}(s, a) - \lambda u(s, a)$, where $\lambda$ is a hyperparameter. We consider the following uncertainty quantification method that uses the maximum learned variance over the ensemble, $u(s, a) = \max_{i=1,...N} ||\Sigma_\theta^i(s, a)||_F$ [57, 22].

As a final process, we train offline RL with the following hybrid objective : $J(\pi) = \frac{1}{1-\gamma} \mathbb{E}_{(I,a) \sim d_f(I,a)}[r(s, a)]$, where $d_f(I, a) = f d_{\mathcal{M}}^{\pi_\beta}(I, a) + (1 - f)d_{\widehat{\mathcal{M}}}^{\eta}(I, a)$, $f \in [0, 1]$ is the

Table 1: Offline RL results for DMControl. The numbers are the averaged normalized scores proposed in [7], where 100 corresponds to expert and 0 corresponds to the random policy. The results with standard deviation are in Appendix **??**.

| Environment | Dataset | IQL | IQL +S2P | CQL | CQL +S2P | SLAC-off | SLAC-off +S2P |
|---|---|---|---|---|---|---|---|
| cheetah, run | random | 10.28 | **12.64**(16.21) | 4.89 | **11.77**(7.52) | 16.37 | **18.14**(35.38) |
| walker, walk | random | -0.28 | **4.03**(0.83) | -0.43 | **10.44**(1.99) | 18.23 | 17.38(20.15) |
| ball in cup, catch | random | 74.77 | **82.28**(80.39) | 84.87 | **92.81**(91.61) | 70.04 | **85.57**(52.81) |
| reacher, easy | random | 33.75 | **70.45**(55.33) | 52.32 | **75.01**(81.48) | 77.43 | **85.76**(87.84) |
| finger, spin | random | -0.17 | **0.46**(-0.11) | -0.01 | -0.11(0.07) | 30.24 | 27.62(32.65) |
| cartpole, swingup | random | 24.52 | **38.59**(29.1) | 27.67 | **32.93**(42.12) | 35.03 | 31.01(52.22) |
| cheetah, run | mixed | 41.68 | **88.53**(70.44) | 92.63 | **93.16**(93.48) | 16.63 | **26.39**(24.42) |
| walker, walk | mixed | 96.07 | 95.49(97.80) | 97.18 | **97.84**(98.70) | 29.02 | **92.60**(67.09) |
| ball in cup, catch | mixed | 41.94 | 37.79(40.65) | 30.82 | **51.28**(37.21) | 28.54 | **40.41**(32.88) |
| reacher, easy | mixed | 66.88 | **75.61**(75.01) | 70.37 | **75.53**(77.54) | 62.49 | **63.59**(77.58) |
| finger, spin | mixed | 98.18 | 94.78(98.65) | 98.54 | 87.17(80.07) | 64.41 | **83.31**(83.29) |
| cartpole, swingup | mixed | 14.49 | 14.04(51.25) | 14.76 | -4.66(36.94) | 14.51 | **16.36**(25.41) |
| cheetah, run | expert | 79.89 | **87.18**(88.89) | 94.20 | **96.28**(95.54) | 8.92 | **14.41**(8.42) |
| walker, walk | expert | 94.34 | **94.97**(94.35) | 95.43 | **97.97**(98.47) | 11.71 | **70.95**(19.66) |
| ball in cup, catch | expert | 28.57 | **28.60**(28.48) | 28.42 | **28.62**(28.68) | 28.56 | **38.87**(28.69) |
| reacher, easy | expert | 52.13 | **58.19**(57.51) | 57.68 | 32.54(48.46) | 26.61 | **42.85**(34.49) |
| finger, spin | expert | 98.42 | 94.42(99.19) | 73.07 | **97.25**(99.51) | 24.75 | **81.05**(52.21) |
| cartpole, swingup | expert | 20.43 | 18.37(18.03) | 19.35 | 18.54(30.22) | 14.11 | 11.18(-3.80) |

ratio of the datapoints drawn from the offline dataset $D$, and $\eta(\cdot|s)$ is the state rollout distribution used with the trained dynamics ensemble model $\hat{T}_\theta$, and $\widetilde{\mathcal{M}}$ is same as $\widehat{\mathcal{M}}$ except that the reward is $\tilde{r}(s,a)$ instead of $\hat{r}(s,a)$. Samples from $d^\eta_{\widetilde{\mathcal{M}}}(I,a)$ can be obtained by rollout $\eta$ in $\widetilde{\mathcal{M}}$ and convert the obtained state transitions $\{(s_t,a_t,r_t,s_{t+1})\}_{t=0}^N$ into $\{(I_t,a_t,r_t,I_{t+1})\}_{t=0}^N$ by using the trained S2P generator in Section 3.1. For implementation, we collect synthetic image transition data in separate replay buffer $D_{model}$ and train the agent by any offline RL algorithms with the sampled mini-batches from $D$ and $D_{model}$ by the ratio of $f$ and $1-f$. The overall algorithm and more training details are summarized in Appendix B.

## 4 Experiments

### 4.1 Environments & Data collection

We evaluate our method on a large subset of the dataset from the DeepMind Control (DMControl) suite [52]. It includes 6 environments, which were typically used for online image-based RL benchmarks. However, to the best of our knowledge, all of these environments have never been properly evaluated in an offline image-based RL setting. The datasets in these benchmarks are generated as follows: **random** : rollout by a random policy that samples actions from a uniform distribution. **mixed** : train a policy using state-based SAC [10] until 500k steps for finger, cheetah and 100k steps for the others, then randomly samples trajectories from the replay buffer. 500k, 100k steps are minimum steps required to reach the expert level performance for each task. **expert** : train a policy using state-based SAC. After convergence, we collect trajectories from the converged policy.

### 4.2 Offline Reinforcement Learning

To validate whether the S2P can help improve the offline RL performance, we evaluate our algorithm with recent offline RL algorithms like **CQL** [26] which utilizes conservative training of the critic, and **IQL** [24] which trains critic by implicitly querying actions near the distribution of the dataset, and **SLAC-off** [32] which is state-of-the-art online image-based RL algorithm. We use this **SLAC-off** with the offline setting. We also compare the policy constraint-based offline RL algorithms like **BEAR** [25], and behavior cloning, **BC**. The results for these two algorithms are in Appendix B. To extend the offline RL into the image-based setting, we follow the image encoder architecture from [32] and train a variational model using the offline data. Then, we train in the latent space of this model.

We include the offline RL results on different types of environments and data in Table 1. The results on the originally given offline dataset (50k) are shown in the left column of each algorithm, and the results on the S2P-based augmented dataset are shown in the right column of each algorithm. The S2P augments the same amount of the original offline dataset (50k). As it physically has more data (100k) than the original dataset (50k), for a reference, we also include the results on the 100k offline dataset in the parenthesis in Table 1. Overall, the S2P-based method achieves better performance than the 50k dataset, even exceeding the 100k dataset's score in some tasks.

As the S2P answers how to generate the image transition data, we also have to consider where to generate the image transition data through S2P. Specifically, we use a random policy as $\eta(\cdot|s)$ in the mixed, expert dataset, and a policy trained by the state-based offline RL as $\eta(\cdot|s)$ in the random dataset. These strategies are considered due to the following two assumptions. Firstly, as the random dataset only has random behavior, the dataset may not have any meaningfully rewarded states even with the augmentation by the random policy, especially in the locomotion environment (e.g. cheetah, walker cannot leave the initial states by the random policy). As the S2P's objective is to extend the distribution of the given dataset, the policy trained in an offline manner can help leave the support of the dataset compared to the naive random policy. Secondly, as the non-random datasets may have relatively biased state-action distributions that receive meaningful rewards compared to the random dataset, it is difficult to get out of the support of these datasets by the trained policy as most of the state-action induced by the trained policy are included in the similar distribution of these datasets. However, the random policy can be effective as it can bring some exploration effects like noise injection or increasing entropy in [35, 10].

To verify such assumptions, we analyze the effect of different types of rollout distribution $\eta(\cdot|s)$ in offline RL performance (Table 2). We denote **50k dataset** as the results of the given original offline dataset, and **+S2P(random $\eta$)** as the results of the data augmented by rollout with the random actions, and **+S2P(offRL $\eta$)** as the results of the data augmented by rollout with the state-based offline RL policy. As expected, the random policy is more effective in the expert dataset. Also, we could find the opposite phenomenon in the random dataset, and trade-offs between these two strategies in the mixed dataset.

| DATASET | METHOD | 50K DATASET | +S2P (RANDOM $\eta$) | +S2P (OFFRL $\eta$) |
|---|---|---|---|---|
| CHEETAH | IQL | 10.28 | -0.107 | **12.64** |
| RUN | CQL | 4.89 | -0.69 | **11.77** |
| RANDOM | SLAC-OFF | 16.37 | 11.94 | **18.14** |
| CHEETAH | IQL | 41.68 | **88.53** | 58.67 |
| RUN | CQL | 92.63 | **93.16** | 89.6 |
| MIXED | SLAC-OFF | 16.63 | 26.39 | **26.53** |
| CHEETAH | IQL | 79.89 | **87.18** | 79.20 |
| RUN | CQL | 94.20 | **96.28** | 93.69 |
| EXPERT | SLAC-OFF | 8.92 | **14.41** | 7.79 |

Table 2: Experiments on each different rollout distribution $\eta(\cdot|s)$ in cheetah-run environment.

## 4.3 Comparison with model-based image RL

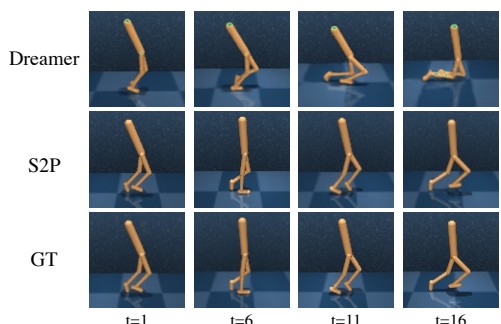

Dreamer

S2P

GT

t=1    t=6    t=11    t=16

Figure 4: Qualitative comparison of S2P and Dreamer.

| dataset | method | 50k dataset | +S2P | +Dreamer |
|---|---|---|---|---|
| cheetah | IQL | 41.68 | **88.53** | 2.09 |
| run | CQL | 92.63 | **93.16** | 58.93 |
| mixed | SLAC-off | 16.63 | **26.39** | 4.68 |
| walker | IQL | 96.07 | **95.49** | 1.28 |
| walk | CQL | 97.18 | **97.84** | 95.88 |
| mixed | SLAC-off | 29.02 | **92.60** | 52.65 |
| cheetah | IQL | 79.89 | **87.18** | 73.23 |
| run | CQL | 94.20 | **96.28** | 53.69 |
| expert | SLAC-off | 8.92 | **14.41** | 3.65 |
| walker | IQL | 94.34 | **94.97** | 34.95 |
| walk | CQL | 95.43 | **97.97** | 96.14 |
| expert | SLAC-off | 11.71 | **70.95** | 52.03 |

Table 3: Quantitative comparison of S2P and Dreamer.

To show why the multi-modal inputs are necessary for augmenting the image transition data in offline image RL, we compare our S2P with the previous model-based image RL algorithm, Dreamer [12], as it can also reconstruct the images of the agent by training the reconstruction error from ELBO objective only using uni-modal inputs (previous images of the agent). For comparison, we trained Dreamer in an offline manner with the same dataset used for training S2P, and predicted future images with the episode context obtained from 5 consecutive ground truth images. Even though the Dreamer saw more previous steps' images compared to S2P (only a single image of the previous

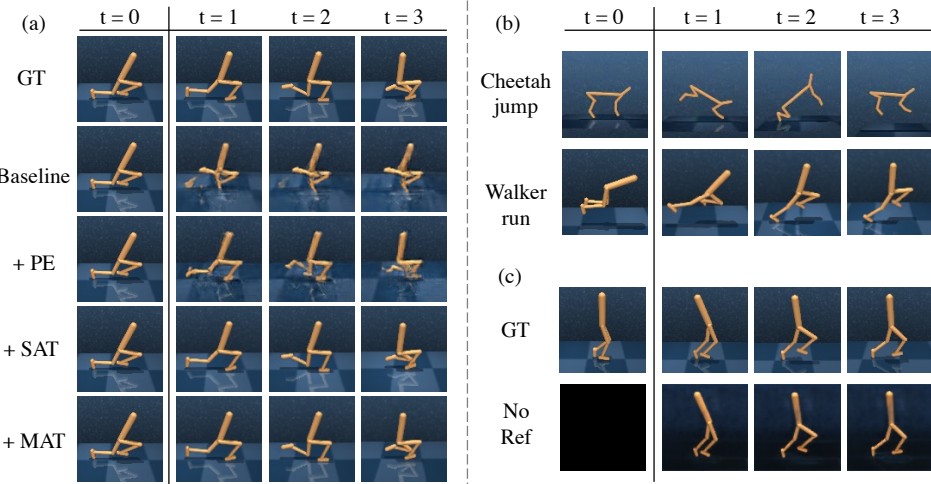

Figure 5: Qualitative results. We report the generator performance by synthesizing multiple steps with a single trajectory. (a) demonstrates **the effectiveness of each component** in the S2P generator where PE, SAT, and MAT indicate positional encoding, state affine transformation and multimodal affine transformation, respectively. SAT cannot perfectly estimate the correct location of the agent as there is **misalignment at the ground checkerboard** compared to MAT. (b) represents **unseen task adaptation** using the trajectories given from the state-level transition model. (c) shows that our model can recover the posture of the agent **without any reference image**. Additional qualitative results on several environments are provided in Appendix A.

step), it still has difficulty in generating accurate posture, which supports the S2P's advantages (Fig 4). It is because the priority of the model-based image RL algorithms is learning the effective latent representation for RL tasks, and they do not utilize a broader source of supervision from the state inputs.

To prove that state-inconsistent images from the model-based image RL method cannot improve offline RL performance at the S2P level, we perform the same experiment in Table 1, but replace the augmented images from S2P with images from Dreamer. We observe that the image augmentation from Dreamer even degrades the original RL performance in several tasks, and the performance improvements with S2P excels the augmentation from Dreamer by a large margin in all baselines (Table 3). Thus, we could say that the inaccurate posture and quality of the images generated by the model-based method trained with the uni-modal inputs (images) are not sufficient for augmenting image transition data in the offline setting. More experiments and details of the augmentation are in the Appendix.

## 4.4 Ablation

To observe how each component of the S2P contributes to the quality of the synthesized images, we perform an ablation study on the model architecture of the S2P and show its qualitative results on Figure 5(a). A baseline architecture without any contribution of our proposed method, i.e. positional encoding (PE) and multimodal affine transformation (MAT), shows the worst image quality. We report that a simple application of the high frequency function $\psi$ to the input state $s_t$ (PE) results in a noticeable increase in image quality. We also address the necessity of the input image $I_{t-1}$ to estimate the modulation parameters, $\gamma$, $\beta$, for the learned affine transformation. We ablate the input $I_{t-1}$ in MAT and utilize only the spatially expanded input state which is called State Affine Transformation (SAT). The generator which replaces the MAT module with SAT has difficulty in exploiting dynamics information from the previous image $I_{t-1}$ and the translation error is accumulated as the generator recurrently synthesizes the long-horizon trajectory. We can observe such dynamics inconsistency especially in the locomotion tasks, e.g. cheetah, and walker, which expresses the velocity in images by the change of the checkerboard in the ground. Compared to SAT, our proposed MAT which leverages both the state $s_t$ and the previous image $I_{t-1}$ shows better performance in reconstructing not only the posture of the agent but also its dynamics-consistent background.

## 4.5 Zero-shot Task Adaptation

To validate whether the S2P can help the offline RL process in settings that require generalization to tasks that are different from the given dataset, we construct two environments **cheetah-jump** and **walker-run**. In **cheetah-jump**, which is referred from [57], the agent should solve a task that is different from the original purpose of the behavior policy. Specifically, we relabel the rewards in the **cheetah-run-mixed** dataset to reward the cheetah to jump as high as possible. Then, we generate the image transition data from the states whose z positions are bigger than a threshold to validate the S2P-based augmentation's advantage in tasks that require generalization. By training with the relabeled reward, the agent with the S2P achieves a higher return and learns to bounce back and forth to take a leap higher (Figure 5(b)), even though the batch data contain little jumping motion.

To investigate whether the S2P can generate unseen image distributions from unseen state distribution, we collect **walker-run** state dataset by the same way of other mixed datasets in Section 4.1. Then, we generate images from these unseen states by recurrently using the S2P generator and we apply offline RL on these generated image transition data. Even though the state distribution is totally different from **walker-walk** dataset as the agent should run instead of walk, not only the S2P successfully generalizes to the unseen image distributions (Figure 5(b)), but also the agent can be trained to run by offline RL only with these synthesized images beyond the expertise of the dataset (Table 4). More details are in Appendix B.

We attribute such satisfactory task generalization to the model architecture of the S2P which leverages both $I_{t-1}$ and $s_t$ for synthesizing the $I_t$. The posture of the agent is deterministic with the sole-state condition and the background of the image such as the ground checkerboard is dependent both on the state and the previous image. Therefore, our S2P exploits the state input to generate the posture of the agent and exploits the image input to generate the background. It is well shown in Figure 5(c) where we intentionally replace all the image input with the zero

| METHOD | WALKER-RUN | METHOD | CHEETAH-JUMP |
|---|---|---|---|
| BATCH MEAN | 572.94 | BATCH MEAN | 2.05 |
| IQL | N/A | IQL | 36.6 |
| IQL+S2P | 659.25 | IQL+S2P | 54.4 |
| CQL | N/A | CQL | 40.8 |
| CQL+S2P | 652.19 | CQL+S2P | 47.2 |

Table 4: Average returns of the walker-run and cheetah-jump tasks. We include mean undiscounted return of the episodes in the batch data for comparison.

matrix $O_{H \times W \times 3}$ during the inference phase. We observe that S2P still perfectly reconstructs the posture of the agent with its corresponding state only while it fails to recover the background and the ground checkerboard as we expected.

## 5   Conclusion

We firstly present the state-to-pixel (S2P) algorithm that synthesizes the raw pixel from its corresponding state and the previous image. As the augmentation paradigm of the S2P is generating dynamics-consistent image transition data, we demonstrate that S2P not only improves the image-based offline RL performance but also shows powerful generalization capability on unseen tasks. Even though S2P shows promising results in offline RL by data augmentation techniques, the assumption that datasets consist of pairs of images and states is still a strong assumption. Thus, for future work, we plan to further extend the idea with more relaxed assumptions such as unpaired datasets or extend other state-based applications to the image-based RL algorithms.

## 6   Acknowledgement

This research was supported by Institute of Information & communications Technology Planning & Evaluation (IITP) grant funded by the Korea government(MSIT) [NO.2021-0-01343, Artificial Intelligence Graduate School Program (Seoul National University)] and by Unmanned Vehicles Core Technology Research and Development Program through the National Research Foundation of Korea(NRF) and Unmanned Vehicle Advanced Research Center(UVARC) funded by the Ministry of Science and ICT, the Republic of Korea(NRF-2020M3C1C1A01086411).

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
