# OpenReview forum: "S2P: State-conditioned Image Synthesis for Data Augmentation in Offline Reinforcement Learning"
_NeurIPS.cc/2022/Conference — NeurIPS 2022 Accept_

### Official Review · Reviewer_WCLw · 2022-07-10

**Rating:** 7
**Confidence:** 4
**Soundness:** 3 good
**Presentation:** 3 good
**Contribution:** 2 fair

**Summary:**

This paper focuses on an offline reinforcement learning problem. A generative model S2P is proposed to generate image transition data by virtually exploring the state space while keeping the dynamics consistency. In detail, a multi-modal affine transformation module is proposed to utilize both state and previous image data to generate modulation parameters. The proposed method is evaluated on the DMControl benchmark, and thorough empirical results are given from aspects including offline policy learning, model-based method, i.e. Dreamer comparison, ablation study, and generalization performance.

**Questions:**

1. Why is high-quality and accurate synthesis image generation important than data generation in latent space? Given the generated synthesis image, the offline agent should also to learn how to encode the image, which may cause additional learning difficulty.
2. What is the offline learning performance if only using state-based affine transformation instead of multimodal affine transformation, i.e. SAT vs. MAT?
3. What is the performance comparison with other offline reinforcement learning method, e.g. COMBO [1], ROMI [2] in which a learned dynamics model are used for data generation?
4. Why use the previous step image instead of the previous step state? By concatenating the current step and previous step state, the features change can also bring velocity which is the mean reason to introduce the image as in line  117:126.
5. Why VGG encoder pre-trained on Imagenet can be used for RL tasks, especially DMControl image observation feature encoding in Eq. 5 since the two domains differ a lot?
6. Is The baselines’ encoder different from S2P as mentioned in line 230:231? How do keep comparison fair?
7. Why introducing more generated image data cannot bring distinct performance improvement as shown in Table 1?

[1] Yu T, Kumar A, Rafailov R, et al. Combo: Conservative offline model-based policy optimization[J]. Advances in neural information processing systems, 2021, 34: 28954-28967.

[2] Wang J, Li W, Jiang H, et al. Offline reinforcement learning with reverse model-based imagination[J]. Advances in Neural Information Processing Systems, 2021, 34: 29420-29432.


Update after rebuttal

All questions have been addressed.

**Limitations:**

This submission is not related to a distinct negative social impact.

**Strengths And Weaknesses:**

**Originality**

Although the model-based generation is commonly-used in reinforcement learning (RL) methods, the image-based offline RL problem is novel, and the authors proposed the first image-based offline data generation method. But the core module in S2P, i.e. the multimodal affine transformation (MAT) is not novel, which has been investigated in previous works [1], [2].

**Quality**

The authors conduct detailed experiments to validate the proposed method S2P, and the results sound. It shows significant performance improvement in 6 image-based DMControl tasks. The authors also explore the influence of different data generation policies and give thorough visualization to help understand the image generation capability and the importance of each module in S2P. The overall empirical results are convincing. But there are still a few problems with the importance of multimodal input and other baselines, which are listed in the following Questions section.

**Clarity**

The paper is well-written and easy to understand. The motivation is clear. Sufficient related literature is included. The experiment configuration details are given in the appendix which helps the further implementation.

**Significance**

The underlying offline RL problem is important both in academic and industrial communities. The authors try to tackle an interesting image-based data generation problem. Although the core module MAT has been investigated in previous computer-vision community, the proposed S2P is still essential since the overall idea is simple and the performance improvement is significant. This encourages the RL community to explore more complex offline RL problems, i.e. with image input.

[1] Li B, Qi X, Lukasiewicz T, et al. Manigan: Text-guided image manipulation[C]//Proceedings of the IEEE/CVF Conference on Computer Vision and Pattern Recognition. 2020: 7880-7889.

[2] Park T, Liu M Y, Wang T C, et al. Semantic image synthesis with spatially-adaptive normalization[C]//Proceedings of the IEEE/CVF conference on computer vision and pattern recognition. 2019: 2337-2346.

---

> ### Author Response · Authors · 2022-08-02
> **Response to Reviewer WCLw (1/2)**
>
> ## Q1. MAT is not novel which is similar to SPADE [1] and ManiGAN [2]:
> We respectfully disagree. As we mentioned in the manuscript (59-61, 87-91), our proposed MAT fuses multi-modal signals in estimating the modulation parameters in Adaptive Instance Normalization (AdaIN) whereas previous methods (SPADE [1], ManiGAN [2]) only utilizes a single domain signal (semantic map, reference image) for style/domain transfer.
>
> Modalities in style transfer module (AdaIN)
>
> **SPADE** [1] input: random variable z, modulation: semantic map
>
> **ManiGAN** [2] input: text, modulation: image
>
> **S2P** input: state and image, modulation: state and image
>
> If there exists other prior literature that also deals with multi-modal input and modulation, we would appreciate it if you could mention it. We will be glad to discuss the difference from S2P.
>
> [1] Li B, Qi X, Lukasiewicz T, et al. Manigan: Text-guided image manipulation[C]//Proceedings of the IEEE/CVF Conference on Computer Vision and Pattern Recognition. 2020: 7880-7889.
>
> [2] Park T, Liu M Y, Wang T C, et al. Semantic image synthesis with spatially-adaptive normalization[C]//Proceedings of the IEEE/CVF conference on computer vision and pattern recognition. 2019: 2337-2346.
>
> ## Q2 : Why is high-quality and accurate synthesis image generation important than data generation in latent space?
> The Dreamer exactly does the data generation in latent space by deploying the latent dynamics. However, the ELBO-based method such as Dreamer only can encode the seen images, which means that the latent feature involves the information of the seen images. That is, if we explore the latent space different from S2P (state space), the visited novel latent feature is not guaranteed to represent the novel state(image) in the original domain. Therefore, it is important to distill as much information as possible into the image encoder of the policy and critic networks by producing diverse image data distribution despite causing an additional learning process to encode the synthesized images.
>
> Furthermore, the advantages of our S2P could be further validated by the generalization test in Section 4.5 for answering the image generation capability at the newly visited states by the learned state dynamics model. For example, the running walker's images could be augmented even though we only have the state transition data and only a single image at t = 0. The Dreamer-like method that generates in latent space cannot adapt to this new task because it has not ever seen the running walker's images.
>
> Also, we could further consider extending the Dreamer with state inputs, as requested by another reviewer (Please refer to the answer of Q3 for the reviewer "bQSP" if you are interested in). But it also has difficulty in reflecting the state information into image generation due to the lack of architectural design for multi-modal inputs.
>
>
> ## Q3 : Offline RL performance with SAT vs MAT
> For the ablation study, we additionally train the generator only using the SAT and tested the offline learning performance in the walker environment. The results are shown in the following Table.
>
> https://drive.google.com/file/d/1T5xV62voEdUM221JzI9Zz-SoF6W8X5DO/view?usp=sharing
>
> As mentioned in Figure 5, misalignment of the checkerboard affects the recognition of the agent’s dynamics and it results in performance degradation.
>
> ## Q4. Performance comparison with COMBO, ROMI
> The referred paper utilizes forward or reverse dynamics for data generation. However, ROMI does not extend the idea into the image domain, and COMBO experimented in the image domain, which uses a latent dynamics model based on ELBO. As far as we know, augmenting the image transition data in COMBO is performed by generating the future image from latent space, and it would still have similar problems observed in Dreamer, because they do not consider the multi-modal inputs (Answered in Q2). The walker environment is also experimented with in COMBO, and it results in a 57.7-76.4 score (Table 2 in COMBO), while our S2P achieves almost expert level (70.95-97.97) score along all baselines.

---

> > ### Comment · Reviewer_WCLw · 2022-08-09
> > **Thanks for response.**
> >
> > Thanks for the response of the above questions. It is now clear about the novelty over previous methods, i.e. SPADE and ManiGAN. The data generation part is also clear.

---

> ### Author Response · Authors · 2022-08-02
> **Response to Reviewer WCLw (2/2)**
>
> ## Q5. Why use the previous step image instead of the previous step state?
> If we do not use the previous image, we cannot accurately predict the image because the state information alone cannot give the background information (e.g. checkerboard on the ground) which contains the velocity information in the image. It can be also validated as shown in Figure 5-c in our work.
>
>
>
> ## Q6. Why VGG model pretrained with ImageNet in RL tasks:
> We do not use the VGG model for RL tasks, but use them for utilizing perceptual loss in training S2P which is for the ‘image generation’ task. In image generation, it is well known that perceptual loss recovers the details of the image such as the fur of the animal, in this case, the tip of the limb [1].
>
> [1] Johnson, Justin, Alexandre Alahi, and Li Fei-Fei. "Perceptual losses for real-time style transfer and super-resolution." European conference on computer vision. Springer, Cham, 2016.
>
> ## Q7. Is baseline encoder different from S2P?
> First, please note that S2P is not an RL algorithm, rather it is an image synthesis model. Additional training images for offline RL are generated from S2P and any image-based offline RL algorithms can take advantage of S2P. During all the experiments, we use the same baseline architecture which means that the encoders are all the same across the offline RL algorithms (IQL, CQL, SLAC-off). Therefore, it is a fair evaluation when we compare offline RL results.
>
> ## Q8. Why introducing more generated image data cannot bring distinct performance improvement as shown in Table 1?
> First, we would like to know if your question means that the scores in baseline +S2P (right column of each baseline offline RL algorithm in Table 1) and scores in parenthesis are not that distinct. Please note that the scores in parenthesis are **NOT** obtained by augmenting more S2P outputs. We are sorry if it was not clear enough.
>
> Specifically, the left column of each baseline(ex: IQL, CQL, SLAC-off) in Table 1 is the trained results with the 50k dataset, and the right column (ex: IQL+S2P, CQL+S2P, SLAC-off+S2P) is the trained results with the original 50k + generated 50k dataset by S2P (totally 100k). And, as we mentioned in our work (Section 4.2), the scores in the parenthesis are the trained results with the 100k ground truth dataset (it does not contain generated data by S2P, and is just additionally collected for reference to match the same amount). We validate through this experiment that RL agents trained with augmented data (ground truth 50k + S2P 50k) produce a comparable or even better performance compared to the agents trained with the same number of ground truth data (ground truth 100k).
>
> If your question was intended to ask for a comparison of results based on the amount of generated data by S2P compared to the originally given offline dataset(50k), we also additionally experimented with the mixing ratios of the datasets. The results are shown in the following Table.
>
> https://drive.google.com/file/d/14J8vgL9wYyrr7Z5s8oFeckTOGO9Gm4MX/view?usp=sharing
>
> We use the originally given 50k offline dataset, and vary the ratio of the generated dataset by S2P. (That is, 5:N means that the ratio of the originally given offline dataset is 5, generated dataset by S2P is N). Except for the already successful baseline such as cheetah-CQL, our S2P overall increases the offline RL performance, regardless of the ratio, in all environments and baselines. Some slightly decreased performances of 5:7.5 compared to the 5:5 or 5:2.5 seem to be the random policy’s effect (As mentioned in our work, the random policy is used for mixed-level datasets), because too much randomness in the dataset could make the overall dataset’s expertise lower.
>
>
> If we misunderstood your question, please kindly let us know to provide proper responses.
>
>
> ## Postscript
> We appreciate all your efforts during the review process. We believe to have addressed all the comments. If this response is not sufficient enough to raise the score, please do not hesitate to let us know.

---

> > ### Comment · Reviewer_WCLw · 2022-08-09
> > **Thanks for your response.**
> >
> > Thanks for your detailed reply. For the Q8, it is the first case, which is clear given the authors explanation. The authors have addressed all my concerns. I have raised my score.

---

> ### Author Response · Authors · 2022-08-08
> **Reminder**
>
> First, we sincerely thank you for reviewing our paper and providing constructive comments with pleasure.
>
> We think we have already addressed issues that you may concern about in our response.
>
> If there are any problems with the attached links, please let us know, and we will address them.
>
> If our response is insufficient for your concerns or there are still several points that are not enough to change your opinion about our work, we politely ask if you could comment on our response with more specific suggestions.
>
> We would really appreciate it if we could have more discussion on our research and it would be helpful for further developing our research towards more valuable works in the offline RL and data augmentation domain.

---

### Official Review · Reviewer_h9Ka · 2022-07-10

**Rating:** 7
**Confidence:** 4
**Soundness:** 3 good
**Presentation:** 3 good
**Contribution:** 3 good

**Summary:**

This work extends the distribution of the data generating policy in offline RL by first learning a generative model (S2P) which synthesizes raw pixels given state information and a previous frame. This generative model is dynamics-consistent and employs multiple modalities which is a new contribution compared to the single modality models of previous works. They show that their data augmentation strategy improves SOTA algorithms like CQL, IQL and an offline version of SLAC. They conduct several ablations showing the value of different components they introduce such as position encoding, multimodal affine transformations, state only affine transformations etc. They also show that even on unseen tasks with unseen image distribution, just having the state distributions allows S2P to generate reliable images to train an offline RL algorithm.


**Questions:**

* I wanted to confirm if the baselines without and with S2P are trained with exactly the same number of transitions. If so, please clarify in the paper. Also, if the total frames are fixed, it will be good to know how the performance varies as a fraction of the mixing ratios.
* There are a few very minor corrections in the paper that can be fixed in the final version if the authors do another pass over the paper  – eg. (“only single the previous image” in Sec 4.3). Section 3.2 claims that “naively optimizing a learned MDP is prone to failure”, but misses appropriate citations.
* There is an interesting ablation where you zero out the image but keep the state intact in Figure 5c. Was the other ablation done ? i.e. to mask the state and only use the image to generate the next image. I understand that you present the Dreamer baseline for this case, but it would be nice to have the ablation on the same model to be more precise about the gains.
* You have shown interesting ablations with SAT, MAT and positional encoding in Figure 5. Since you introduce three components of the loss, it will also be interesting to see what value each component adds in terms of generalization.


**Limitations:**

The authors have not addressed any specific limitations of the work. Please refer to the earlier replies for potential areas of improvement.

**Strengths And Weaknesses:**

This paper proposes a state-to-pixel generative model that can be used to generate dynamics-consistent transition augmentations to any offline RL algorithms. The paper is clearly written, describes the contributions and how it differs from other similar works in the area. The paper also conducts extensive experiments with three different algorithms on the DMControl datasets. The results are solid and the ablations show the values of each of the algorithmic pieces that were added. I believe that the work is still significant and demonstrates that augmenting data with dynamics-consistency is significantly useful for any offline RL learning algorithm. Thus it is a useful contribution to the offline RL research.

The paper has conducted experiments limited to only DMControl with reliable state transitions. It would make the model claims stronger if one were to see if the same data augmentation strategy could be applied to other image observation environments and partially observable environments. This model relies on the fact that the state representations are reliable and learns to generate new transitions via “exploring” in the state space and generating corresponding image observations. Not all problems provide reliable state transitions and representations and this work has not fully explored cases where state representations/dynamics are intrinsically less reliable and noisy.

---

> ### Author Response · Authors · 2022-08-02
> **Response to Reviewer h9Ka (1/2)**
>
> ## Q1. S2P with noisy or unreliable state transition
> As you mentioned, we expect S2P not to perform reasonably where the state transition is not reliable or noisy (e.g. extremely complex dynamics of the agent, or real-world data with sensor noise). Nevertheless, we believe that the state transition is still much more refined and reliable information compared to pixel-level transition data, and therefore, leveraging S2P with noisy state transition is still a good choice for data augmentation in offline image RL settings.
>
> ## Q2. The size of the dataset for training,  Mixing ratio
> First of all, we would like to make clear that we did not use any additional ground truth data when we obtain all the offline RL results of +S2P experiments in our work including Table 1. We only use the originally given fixed 50k dataset to train S2P, and any additional data used for obtaining the results in +S2P experiments is only generated by the S2P (It is not ground truth data). Therefore, it could be considered fair even though the total number of transitions is different when we address the data augmentation problems.
>
> Specifically, the left column of each baseline(ex: IQL, CQL, SLAC-off) in Table 1 is the trained results with the 50k dataset, and the right column (ex: IQL+S2P, CQL+S2P, SLAC-off+S2P) is the trained results with the original 50k + generated 50k dataset by S2P (totally 100k). And, as we mentioned in our work, the scores in the parenthesis are the trained results with the 100k ground truth dataset (it does not contain generated data by S2P, and is just additionally collected for reference to match the same amount). We validate through this experiment that RL agents trained with augmented data (ground truth 50k + S2P 50k) produce a comparable or even better performance compared to the agents trained with the same number of ground truth data (ground truth 100k).
>
>
> Therefore, we think fixing the number of entire transition data is not a proper experiment setting for evaluating the performance according to the ratio of ground truth and synthesized frames.
> Rather, we fix the number of the ground truth and change the number of the synthesized images to figure out how much additional data effects the RL performance.
> The results are shown in the following table.
>
> https://drive.google.com/file/d/14J8vgL9wYyrr7Z5s8oFeckTOGO9Gm4MX/view?usp=sharing
>
> We use the originally given 50k offline dataset, and vary the ratio of the generated dataset by S2P. (That is, 5:N means that the ratio of the originally given offline dataset is 5, generated dataset by S2P is N). Except for the already enough successful baseline such as cheetah-CQL, our S2P overall increases the offline RL performance, regardless of the ratio, in all environments and baselines. Some slightly decreased performances of 5:7.5 compared to the 5:5 or 5:2.5 seem to be the random policy’s effect (As mentioned in our work, the random policy is used for mixed-level datasets), because too much randomness in the dataset could make the overall dataset’s expertise lower.

---

> > ### Comment · Reviewer_h9Ka · 2022-08-09
> > **Thanks for the comments and the additional ablations -- they make the paper more thorough.**
> >
> > Thanks a lot for the comments
> >
> > Q2. Thanks for the detailed explanation. I have no further questions on the amount of data used. Thanks also for the demonstration on mixing different amounts of augmented data. I hope you are able to include this in the camera-ready version.
> >
> > Q4. The additional ablation looks good. It is convincing to know that masking out position and velocity provides the agents capabilities that we expect them to see. Thanks for the detailed explanation.
> >
> > Q5. Thanks for the ablation. Kindly try to include this in the draft.
> >
> > After the additional ablations and explanations, I am increasing the score to a 7 still noting the limitaton highlighted originally.

---

> ### Author Response · Authors · 2022-08-02
> **Response to Reviewer h9Ka (2/2)**
>
> ## Q3. Typos & Reference
> We fix typos and cite missing references in the newly uploaded manuscript file (line 196, 273).
>
> ## Q4. Ablation on state masking:
> We appreciate your brilliant idea, and we additionally experiment with ablation studies that mask out the state information.
>
> https://drive.google.com/file/d/1-nnZCoFpbMKGVllnQc6LIdl7Bntgk-F2/view?usp=sharing
>
> We mask out either the position or velocity of the state and each presents different aspects. When we mask out the position of the state, S2P cannot capture the posture of the agent, but it captures the velocity by generating the change of the backgrounds (checkerboard ground) as we expected. When the velocity of the state is masked out, S2P still recovers the posture of the agent, but it walks in place without moving forward due to the missing velocity.
> Without the previous image, S2P recovers the posture of the agent plausibly, but as it cannot access the previous image, the background cannot be reconstructed.
>
> ## Q5. Ablation on the loss function
> All the loss combined shows better qualitative results compared to the sole L1 loss as shown in the figure at the below link.
>
> https://drive.google.com/file/d/1Ef7ahMfcikyJhnpX7bbWKOfQKtucKxZd/view?usp=sharing
>
> We attribute such image quality to the adversarial GAN loss with perceptual loss as GAN generates photorealistic images and perceptual loss is known as synthesizing the details of the image [1] such as hair or fur, in this case, the tip of the limb.
>
>
> [1] Johnson, Justin, Alexandre Alahi, and Li Fei-Fei. "Perceptual losses for real-time style transfer and super-resolution." European conference on computer vision. Springer, Cham, 2016.
>
> ## Postscript
> We appreciate all your efforts during the review process. We believe to have addressed all the comments. If this response is not sufficient enough to raise the score, please do not hesitate to let us know.

---

> ### Author Response · Authors · 2022-08-08
> **Reminder**
>
> First, we sincerely thank you for reviewing our paper and providing constructive comments with pleasure.
>
> We think we have already addressed issues that you may concern about in our response.
>
> If there are any problems with the attached links, please let us know, and we will address them.
>
> If our response is insufficient for your concerns or there are still several points that are not enough to change your opinion about our work, we politely ask if you could comment on our response with more specific suggestions.
>
> We would really appreciate it if we could have more discussion on our research and it would be helpful for further developing our research towards more valuable works in the offline RL and data augmentation domain.

---

### Official Review · Reviewer_bQSP · 2022-07-11

**Rating:** 5
**Confidence:** 4
**Soundness:** 3 good
**Presentation:** 3 good
**Contribution:** 2 fair

**Summary:**

This paper proposes S2P: a method for state-conditioned image synthesis for offline image-based RL. The authors motivate their method by the fact that offline RL suffers from distribution shifts in the low data regime, and augmenting datasets using a learned dynamics model has previously been shown to be effective for mitigating this distribution shift. The proposed method is evaluated on 6 continuous control tasks from DMControl by training a number of offline RL methods on the original dataset as well as an augmented dataset and comparing the difference in downstream performance.

**Questions:**

I believe that my greatest concerns are already made explicit in the "weaknesses" section above. Specifically, I'd like the authors to

- Explicitly state key assumptions made + limitations of the method
- Justify why simpler approaches are not sufficient given the assumption of (I,s) pairs in the dataset
- Clarify whether Dreamer leverages the available state information in the comparison, and if not, what the justification for that is

**Limitations:**

The authors do *not* discuss potential negative societal impact of their work and do *not* explicitly discuss limitations of their method despite claiming so in the checklist. See my previous comments on limitations. I do not anticipate ethical issues.

**Strengths And Weaknesses:**

Strengths:
- Simple and intuitive extension of a state-based RL method to image-based RL.
- The paper is generally easy to follow and the contributions are clearly presented.
- Experimental evaluation demonstrates that synthetic transitions are valuable across a variety of algorithms, tasks, and data distributions. The comparison to conventional image augmentations in appendix is appreciated.

Weaknesses:
- The assumption that datasets consist of pairs of images and ground-truth states is obviously a very strong assumption that will not hold in a lot applications. This is not a problem in itself, but the authors don't seem to ever state it explicitly in the paper. I'd like to see this made absolutely unambiguous to readers.
- Given the above assumption of (I,s) pairs, there are a number of conceptually simpler methods that may or may not suffice. For example, one could simply learn an I -> s predictor using supervised learning and then apply one of the existing state-based offline RL algorithms. A slightly more sophisticated method could learn the I -> s predictor, augment the dataset using a state-based dynamics model, and then learn a state-based policy on the augmented dataset, and several other such approaches come to mind. In light of this, I'd like the authors to justify their more complex approach, ideally backed by data although that may not be feasible during the rebuttal itself.
- It is unclear if it is indeed the case, but it appears that Dreamer does not leverage state information at all in the comparison, although it is assumed available. A more fair comparison would be to also condition the RSSM of Dreamer on state. Judging by the results in appendix, it appears that data quantity is not a big factor in the poor performance of Dreamer, so such an experiment would help clarify whether the poor results indeed are due to the objective or lack of access to privileged state information.

**Post-rebuttal:** The author response addresses my main concerns and I am willing to raise my rating with the expectation that authors include reviewer suggestions in a future revision.

---

> ### Author Response · Authors · 2022-08-02
> **Response to Reviewer bQSP (1/2)**
>
> ## Q1. Explicitly mention limitation
> Training S2P requires an assumption that state and its paired images are needed, so we explicitly mention it in the manuscript. Please check the newly uploaded manuscript (340-343).
>
> ## Q2. Image to state (P2S) vs state to image (S2P)
> Thank you for your interesting idea. To address your comment, we additionally experiment with the image-to-state model (P2S) compared to S2P.
> We train the state estimator (P2S) in two different ways: 1) train state estimator from a single image, 2) train state estimator from three consecutive stacked images. The difference is that we induce the latter one to capture the velocity information from the stacked images as previous image-based RL methods do. The state estimator network consists of CNN layers followed by MLP layers and it is trained by MSE loss between the predicted states and ground truth states. Then, we train a state-based offline RL with the augmented dataset, where the states in the dataset are predicted by the trained state estimator network. And we evaluate the performance by sampling the action from the state-based policy, where the state input for the policy is predicted from the image given by the environment at every timestep. The results are shown in the following Table.
>
> https://drive.google.com/file/d/16JSTfPjjQhNNsksagCpgphGxpSsv81Vs/view?usp=sharing
>
> As we expected, the state estimator from stacked images produces better performance compared to the state estimator from a single image. However, they cannot outperform S2P in any of the tasks, and even cannot outperform the baseline algorithm without S2P in most of the tasks.
> The reason for such performance degradation is that it is difficult to estimate the accurate state information from images that lie in a high-dimensional space. Also, even though the MSE loss is decreased on the given paired image-state dataset by overfitting, the agent observes the unseen image during evaluation, which leads to inaccurate state estimation from unseen image input, and results in drastic performance degradation.
>
> Unlike the state estimation strategy (P2S), S2P leverages multi-model inputs (state, previous image) which have similar dimensionality to the outputs, and it makes S2P synthesize accurate images extracting information from two different modalities. Also, we adopt three different loss functions (L1, GAN, perceptual) so that the combination of the losses mitigates the overfitting to the offline dataset’s distribution.

---

> ### Author Response · Authors · 2022-08-02
> **Response to Reviewer bQSP (2/2)**
>
> ## Q3. State and Image conditioned Dreamer vs S2P
> As you thought that it is a fair evaluation for Dreamer to train with the state information same as S2P, we trained the image and state concatenated Dreamer in cheetah, walker environments. Specifically, the encoder of Dreamer takes the image and state, and the decoder reconstructs the image and state. The state reconstruction loss is the same as the image reconstruction loss (likelihood maximization). Then, we augment the dataset by generating image transition data the same as +DREAMER case in Table 3 in our work (or Table 10 in the appendix). The offline RL performance of state concatenated Dreamer is shown in the following Table.
>
> https://drive.google.com/file/d/1fDN3mS4kxAvnbf_AoP2VSq9ufrZe1JvK/view?usp=sharing
>
> The state-concatenated Dreamer usually (9 of 12) shows better performance compared to the naive Dreamer. But, it still produces worse performance in all the tasks compared to S2P. It is because the architecture of Dreamer itself cannot capture the multi-modal input effectively, while we design S2P for leveraging multi-modal inputs to synthesize image (MAT) and S2P leverages both state and image information for predicting the agent’s image. To visualize the difference between S2P and Dreamer in exploiting multi-modality, we mask out the state and see how the output image is affected. When we zero-mask out the state in Dreamer, there is no significant difference.
>
> In case of S2P, on the other hand, when we remove the position element of the state, S2P does not produce agents, but only reconstructs the backgrounds which contain the velocity information of the state. When we zero-mask out the velocity information in S2P, the agent walks but it walks in the place without moving forward as the velocity is zero.
>
> state concatenated dreamer- walker:
> https://drive.google.com/file/d/146qw83A3S-m1-MHTE8xxLeyp8jeF1Bsr/view?usp=sharing
>
> state concatenated dreamer- cheetah:
> https://drive.google.com/file/d/1cMB903IUdApNRJE0Tf09Xz2PHItNdizh/view?usp=sharing
>
> S2P position/velocity zero-masking :
> https://drive.google.com/file/d/1-nnZCoFpbMKGVllnQc6LIdl7Bntgk-F2/view?usp=sharing
>
>
> ## Q4. Societal Impact
> This work inherits, if any, the potential negative societal impacts of reinforcement learning. We do not anticipate any additional negative impacts that are unique to this work.
>
> ## Postscript
> We appreciate all your efforts during the review process. We believe to have addressed all the comments. If this response is not sufficient enough to raise the score, please do not hesitate to let us know.

---

> > ### Comment · Reviewer_bQSP · 2022-08-04
> > **Response to Authors**
> >
> > Thank you for the comprehensive response. I appreciate the additional justification, as well as both qualitative and quantitative experiments. I agree with the authors' assessment that the actual performance of these additional baselines (P2S in particular) are highly dependent on the particular implementation details, but I believe that the surrounding discussion helps motivate + position the proposed method wrt. other obvious (albeit naive) approaches. I understand that a rebuttal period does not leave sufficient time for additional (rigorous) experiments, so I suggest the authors spend a little more effort post-rebuttal to present P2S fairly (e.g., simple measures to mitigate overfitting if the authors identify that as a cause for poor performance). Lastly, I also found the qualitative masking experiments insightful -- they confirm a suspicion that many readers will probably have. I suggest you include these additional experiments in future revisions along with the justifications for your method that you detailed in your response.
> >
> > Overall, I believe that your response addresses my concerns wrt. justification of approach and am willing to raise my rating.

---

> > > ### Author Response · Authors · 2022-08-06
> > > **Additional experiment for regularizing P2S training**
> > >
> > > Thank you for your constructive review, and we appreciate your suggestion for additional experiments on regularization effects on P2S performance.
> > >
> > > We did two different experiments to verify that
> > > 1) poor performance in P2S is really due to overfitting.
> > > 2) simple regularization (batch/layer normalization) can alleviate such performance drop.
> > >
> > > https://drive.google.com/file/d/182etPm6zZaTfOpi0Yl4J4UNQEiLF6T2A/view?usp=sharing
> > >
> > > For the first experiment, we fix the offline state-based RL model trained with the state transitions generated from P2S and evaluate it with the ground truth state from the environment.
> > > If the model performs well on the GT state during evaluation, it means that P2S predicts states well during training. Otherwise, if the model does not perform well during the evaluation, it indicates failures in generalization on estimating states from the unseen images.
> > >
> > > As we expected, evaluation with GT state (**GT state** column in the table) outperforms evaluation with P2S on the walker-walk dataset.
> > > On the other hand, there is no significant difference between GT state and P2S on the cheetah-run dataset (both GT and P2S produce poor performance).
> > >
> > > The results show that P2S predicts the state well from the image during P2S training on the walker-walk dataset, but it fails to generalize in the unseen image (transition).
> > > Also, P2S cannot even estimate the state well during training on the cheetah-run dataset.
> > > It indicates that the performance of P2S is inconsistent across various environments.
> > >
> > > For the second experiment, we add batch/layer normalization (**BN/LN**) in the convolution layer, which is known as effective in the regularization with image input.
> > > We observe that such a regularization strategy (especially LN) produces much better performance compared to outputs without regularization (However, it still cannot outperform S2P).
> > >
> > > Through two experiments, we conclude that the poor performance in P2S is due to overfitting, and some regularization can alleviate such overfitting with the normalization layer in the models.
> > >
> > > Still, P2S produces inconsistent performance across the tasks and still shows poor performance compared to S2P.
> > > It justifies the need for S2P as a data augmentation strategy in offline image-based RL algorithms.
> > >
> > > We really appreciate your advice, and if there is an additional experiment you want, we will do our best to fulfill your standard.
> > >
> > > Thank you.

---

> > > > ### Comment · Reviewer_bQSP · 2022-08-07
> > > > **Acknowledgement**
> > > >
> > > > I appreciate the additional results and discussion surrounding this important baseline, as well as your thorough discussion with other reviewers. Including all of this reviewer feedback will definitely strengthen your arguments. I already updated my score and am leaning slightly towards acceptance. Your method makes very strong (and perhaps not very realistic given the tasks) assumptions but I believe that the insights will be valuable to the community. Perhaps it would be a good idea (in a future revision perhaps) to put more emphasis on domains where access to (partial) state is a reasonable assumption, e.g. robotics where access to proprioceptive state + camera feedback is commonplace.

---

### Official Review · Reviewer_i7hg · 2022-07-12

**Rating:** 5
**Confidence:** 4
**Soundness:** 3 good
**Presentation:** 4 excellent
**Contribution:** 3 good

**Summary:**

This paper suggests image synthesis method to improve data efficiency in offline RL.  In offline RL, we can only use stored data, but it is too small for improving performance. Thus, many researches try to learn model dynamics, and explore new states. For image based RL, they need observations as pixels. Thus, authors generate image data from new states with recurrent generative model from input state and previous observed image.

**Questions:**

1. Standard deviation of scores in tables.
2. In table1, they mix dataset with random and expert dataset. Then, in some tasks, mixed dataset improve performance rather than expert dataset. It means that expert dataset cannot show diverse state and random policy would be Thus, S2P can be not needed, and we can improve performance with simple augmentations or exploration with random policy or small perturbed policy.
3. How much do you need data to train S2P model?
4. How can you evaluate the quality of synthesized images?

**Limitations:**

It is hard to check the quality of synthesized images without a proper metric for evaluation.
Now, I cannot assure that this generative model is needed instead of other data augmentation methods without comparison.
If S2P does not improve performance significantly, it is not used because of the load of training for generative model.

**Strengths And Weaknesses:**

Strength
To generate data in offline RL, they try to use multi-modal generative model for enough information.
S2P largely improves the performance in DMControl tasks.

Weakness
Not enough baseline, there are no comparison with other augmentation.

---

> ### Author Response · Authors · 2022-08-02
> **Response to Reviewer i7hg**
>
> ## Q1 : Standard deviation of scores in tables.
>
> We attached the standard deviation of the tables.
>
> Table 1 : https://drive.google.com/file/d/1OnymWPD1R_r_1fB3zcE0evagX9oyGSFr/view?usp=sharing
>
> Table 2 : https://drive.google.com/file/d/1hY5wW63VRHKd8UH0W6K6UaDwn6yU_cpD/view?usp=sharing
>
> Table 3 : https://drive.google.com/file/d/1YmdM01Q_bNG6bna4qphX2UCB8ojJ9kfU/view?usp=sharing
>
> Table 4 : https://drive.google.com/file/d/11M_etepameN7zoU2fmpmh1_ECTUrjk-H/view?usp=sharing
>
> ## Q2: S2P vs Naive image augmentation
> S2P outperforms the previous most frequently used & effective image augmentation strategies, REFLECT&CROP, for image RL. Please refer to Table 10 in Appendix. (This strategy’s effectiveness compared to other augmentation strategies like convolution, color-jittering, etc, is already proved experimentally in several model-free image RL papers such as CURL[1], RAD[2], DrQ[3]).
>
> It is because a simple augmentation strategy is just image manipulation rather than broadening the dataset distribution’s support as mentioned in our work.
>
> [1] : Srinivas, Aravind, Michael Laskin, and Pieter Abbeel. "Curl: Contrastive unsupervised representations for reinforcement learning." arXiv preprint arXiv:2004.04136 (2020).
>
> [2] Laskin, Misha, et al. "Reinforcement learning with augmented data." Advances in neural information processing systems 33 (2020): 19884-19895.
>
> [3] Yarats, Denis, Ilya Kostrikov, and Rob Fergus. "Image augmentation is all you need: Regularizing deep reinforcement learning from pixels." International Conference on Learning Representations. 2020.
>
> ## Q3: S2P is not needed with random policy?
>
> Q3.1: First of all, please note that S2P is not some kind of RL algorithm, as mentioned in the "General Comments" above.
>
> Q3.2: If you meant "S2P is not necessary because we can improve the performance with random policy", then S2P is needed anyway to leverage random or perturbed policy in image RL by synthesizing images from the randomly visited state.
>
> If we misunderstood your question, please rephrase it or let us know.
>
>
> ## Q4. How much data do you need to train S2P model?
> The same amount of data used for training the offline RL algorithm is used to train S2P, which is 50k in this work.
>
> ## Q5. How can you evaluate the quality of synthesized images?
> We evaluate the image fidelity of our S2P on 4 different metrics (FID, LPIPS, PSNR, SSIM)  in Table 6 (Appendix) which are widely adopted to validate the quality of generated images. Our S2P outperforms Dreamer in **all** the metrics on average.
>
> ## Postscript
> We appreciate all your efforts during the review process. We believe to have addressed all the comments. If this response is not sufficient enough to raise the score, please do not hesitate to let us know.

---

### Author Response · Authors · 2022-08-02
**General Comments**

We sincerely thank all the reviewers for providing constructive comments.

1. We provide a correction for common misunderstandings of the reviewers to clarify our methodology.
**S2P itself is not an RL algorithm.** Rather, it is an image synthesis algorithm for data augmentation. S2P only generates images from states in the extended distribution and the previous image. The synthesized image outputs can be used as additional data during training for any image-based offline RL algorithms.

2. We politely ask if the reviewers could check the **Appendix** in the supplementary files. Most of the answers are already provided in it.

3. We have uploaded a new version of the manuscript based on reviewers’ comments and minor corrections.

---

### Meta-Review · Area_Chair_w3t2 · 2022-08-29

**Recommendation:** Accept
**Confidence:** Certain

**Metareview:**

## Summary
The performance of the offline RL methods can be limited by the amount of coverage in the dataset. Most real-world problems have limited coverage in the offline RL datasets and the sample efficiency of the pixel-based offline RL methods in general is often poor. Thus, it is an important direction of research to improve the sample-efficiency those continuous control pixel-based offline RL algorithms. This paper proposes a method called S2P which generates pixel based observations from the states by using a generative model. The paper shows improved results on offline DeepMind control datasets.

## Decision

The paper in general is well-written and clear. The idea is simple and seems to be effective compared to other data augmentation approaches. The reviewers were in general positive about this paper. I think the NeurIPS and offline RL community *would benefit from the findings of this paper*.  However, I think a few clarifications in the final version of the paper would make the contributions of this paper more clear.

1. I found the improvements shown in the paper very encouraging. However, I found the choice of the dataset odd and confusing. In particular, I am curious why the authors did not decide to use the standards datasets published in RL Unplugged benchmark for offline RL. I think the authors should justify why they did not use those datasets in the camera-ready version of the paper, provide results on those datasets, and release the datasets that they used in this paper (perhaps contacting the RL Unplugged authors to see if it is possible to release them under RL Unplugged benchmark.) As it stands out, this paper only compares against baselines that the authors themselves implemented in the paper. Nevertheless, running experiments on the RL Unplugged would enable us to be able to compare S2P against other published offline RL baselines.
2. Authors should include the standard deviations in the camera-ready version of the paper as request by *reviewer i7hg*.
3. In general, I think the authors did a good job during the rebuttal and many of the reviewers raised their scores as a result of additional results and experiments that the authors have provided. The authors should include those results in camera-ready version of the paper including the clarifications about the questions that the reviewers asked in the rebuttal.
4. Currently the links and the references in the supplementary material are all broken. The authors should fix those in the camera-ready version of the paper.

**With the above points addressed in the camera-ready version of the paper, I think this paper would be ready for publication.**


**Award:**

No

---

### Decision · Program_Chairs · 2022-09-14

Accept